# Fibre Laser Treatment of Beta TNZT Titanium Alloys for Load-Bearing Implant Applications: Effects of Surface Physical and Chemical Features on Mesenchymal Stem Cell Response and *Staphylococcus aureus* Bacterial Attachment

**Clare Lubov Donaghy [1],\*, Ryan McFadden [1], Graham C. Smith [2] , Sophia Kelaini [3] , Louise Carson [4], Savko Malinov [1], Andriana Margariti [3] and Chi-Wai Chan [1],\***

[1] School of Mechanical and Aerospace Engineering, Queen's University Belfast, Belfast BT9 5AH, UK; rmcfadden02@qub.ac.uk (R.M.); S.Malinov@qub.ac.uk (S.M.)

[2] Department of Natural Sciences, University of Chester, Thornton Science Park, Chester CH2 4NU, UK; graham.smith@chester.ac.uk

[3] Centre for Experimental Medicine, Queen's University Belfast, Belfast BT9 7BL, UK; s.kelaini@qub.ac.uk (S.K.); a.margariti@qub.ac.uk (A.M.)

[4] School of Pharmacy, Queen's University Belfast, Belfast BT9 7BL, UK; l.carson@qub.ac.uk

\* Correspondence: cdonaghy38@qub.ac.uk (C.L.D.); c.w.chan@qub.ac.uk (C.-W.C.)

**Abstract:** A mismatch in bone and implant elastic modulus can lead to aseptic loosening and ultimately implant failure. Selective elemental composition of titanium (Ti) alloys coupled with surface treatment can be used to improve osseointegration and reduce bacterial adhesion. The biocompatibility and antibacterial properties of Ti-35Nb-7Zr-6Ta (TNZT) using fibre laser surface treatment were assessed in this work, due to its excellent material properties (low Young's modulus and non-toxicity) and the promising attributes of fibre laser treatment (very fast, non-contact, clean and only causes changes in surface without altering the bulk composition/microstructure). The TNZT surfaces in this study were treated in a high speed regime, specifically 100 and 200 mm/s, (or 6 and 12 m/min). Surface roughness and topography (WLI and SEM), chemical composition (SEM-EDX), microstructure (XRD) and chemistry (XPS) were investigated. The biocompatibility of the laser treated surfaces was evaluated using mesenchymal stem cells (MSCs) cultured in vitro at various time points to assess cell attachment (6, 24 and 48 h), proliferation (3, 7 and 14 days) and differentiation (7, 14 and 21 days). Antibacterial performance was also evaluated using *Staphylococcus aureus* (*S. aureus*) and Live/Dead staining. Sample groups included untreated base metal (BM), laser treated at 100 mm/s (LT100) and 200 mm/s (LT200). The results demonstrated that laser surface treatment creates a rougher (Ra value of BM is 199 nm, LT100 is 256 nm and LT200 is 232 nm), spiky surface (Rsk > 0 and Rku > 3) with homogenous elemental distribution and decreasing peak-to-peak distance between ripples (0.63 to 0.315 μm) as the scanning speed increases (p < 0.05), generating a surface with distinct micron and nano scale features. The improvement in cell spreading, formation of bone-like nodules (only seen on the laser treated samples) and subsequent four-fold reduction in bacterial attachment (p < 0.001) can be attributed to the features created through fibre laser treatment, making it an excellent choice for load bearing implant applications. Last but not least, the presence of TiN in the outermost surface oxide might also account for the improved biocompatibility and antibacterial performances of TNZT.

**Keywords:** mesenchymal stem cell (MSC); antibacterial performance; TNZT; beta titanium; fibre laser treatment

## 1. Introduction

Aseptic loosening is the most commonly cited indication of load bearing orthopaedic implant revision surgeries [1]. A mismatch in elastic modulus of bone and implant means that required stresses for bone remodelling are not obtained, leading to stress shielding which causes bone resorption and ultimately results in aseptic loosening. The ability of an implant to successfully integrate into native tissue is determined by the surface features, namely surface roughness and topography (physical), wettability (physiochemical) and chemistry (chemical). All of these play a crucial role in modulating cell–surface interactions. The ideal surface conditions for optimal osseointegration still remain to be fully elucidated, however, a consensus exists that surface features are vital in the control of cell response (adhesion, proliferation and differentiation). Among several cell types involved in the osseointegration process, mesenchymal stem cells (MSCs) are multipotent progenitor cells that are responsible for self-replicating and differentiating into varying lineages. The osteogenic lineage includes two fundamental cell types, osteocytes and osteoblasts, which are responsible for the formation of new bone. The hematopoietic-derived osteoclasts resorb bone cells and are crucial for bone homeostasis [2]. Insufficient mechanical loads at the implant site lead to exacerbated osteoclast activity [3], which can potentially be reduced by choosing a material with lower elastic modulus, a property belonging to beta (β) titanium (Ti) alloys.

Ti-based alloys, namely commercially pure (cp) Ti and Ti-6Al-4V, have been successfully applied for orthopaedic applications in the past few decades on account of their promising mechanical and corrosion properties, as well as desirable biocompatibility. In principle, they are classified as alpha (α), near-α, α + β, metastable β or stable β, depending upon material composition (α or β-stabilizers) and thermo-mechanical processing history. β-stabilizers, such as Nb and Ta, are isomorphous, while Zr is a neutral stabilizer [4]. β Ti alloys offer unique characteristics in comparison with their cp Ti (α Ti) and Ti-6Al-4V (α + β Ti) counterparts. Among a family of β Ti alloys which have been developed for orthopaedic applications in recent years, the Ti-Nb-Zr-Ta (TNZT) quaternary alloy system is particularly promising due to its excellent properties, which include superior biocompatibility, low elastic modulus (55 GPa) [5], good corrosion resistance and the absence of toxic elements such as aluminium (Al) and vanadium (V). The cytotoxicity and adverse tissue reaction caused by Al and V have been widely reported in literature [6–9]. Another notable flaw in the choice of the conventional material used for load-bearing orthopaedic implants is the staggering difference in elastic modulus of cortical bone (7–30 GPa) [10] and Ti-6Al-4V (110 GPa) [11]. The utilisation of TNZT can aid in reducing aseptic loosening and stress shielding, and subsequently can cause a reduction in tissue reaction to particulate debris, while surface modification in the form of fibre laser treatment can be implemented to reduce bacterial infection and improve native cell adhesion.

Bacterial infection caused by *Staphylococcus aureus* (*S. aureus*) is considered to be a major issue with this particular strain, accounting for the majority (34%) of implant associated infections [12]. The process is initiated through three core steps: bacterial adherence to the implant surface, bacterial colonisation and lastly biofilm formation. The biofilm consists of layers of bacteria covered in a self-produced extracellular polymeric matrix, hence phagocytosis cannot occur and antibiotics have no effect [13]. Bacteria are non-specific with their adherence, attaching to both rough and smooth surfaces, and to different types of materials. Bacterial adherence and subsequent biofilm development is detrimental to the performance of implants, and the ensuing infection can also be a cause of significant morbidity and mortality to implant patients. Therefore, implementing strategies to minimise the likelihood of initial bacterial adherence to the implant surfaces is crucial to prevent bacterial infection [14,15].

Findings in recent research [16,17] shows that surface modification by laser treatment can be used to reduce bacterial infection and improve native cell adhesion. Laser surface treatment, particularly when carried out by fibre laser technology, is a novel technique for implant surface modification [18–20], providing a clean, fast and highly repeatable process. One important characteristic of the laser surface treatment implemented in this study is that the rapid solidification process produces a homogenous

surface with little thermal penetration, resulting in little to no distortion [21]. Laser treatment has been shown to improve surface hardness and wear corrosion [22–25]. Other sources of pain after primary total hip replacement (THR), as recorded by the UK National Joint Registry (NJR), include adverse soft tissue reaction to particulates and to infection, both of which can be improved upon by combining better material selection in the form of a novel beta titanium alloy with surface modification using fibre laser treatment. In terms of the economic benefit, TNZT is potentially superior to currently used materials, as it, coupled with laser surface treatment, can simultaneously decrease the most common indications for hip revision surgery listed in the NJR reports [1,26–29]. TNZT use for hip stems may also have a longer life span beyond the current three quarters of hip replacements that last 15–20 years [30] before a revision surgery is required. A systematic review conducted suggested that just over half of hip replacements last 25 years [30]; as ever, there is room for improvement.

If a quantitative relationship between the surface features of an implant and the cell responses were to be established, implants could be designed with specific surfaces which would aid in the host's natural healing processes [31]. To date, very little work has been conducted using the aforementioned composition of beta titanium alloy, and even less specifically focusing on the effect of surface features in relation to MSC response.

The study objectives were to investigate surface feature effects, namely, roughness, topography, composition and chemistry of untreated and fibre laser treated beta titanium alloy Ti-35Nb-7Zr-6Ta on human MSC response by assessing attachment, proliferation and differentiation at various time points, as well as *S. aureus* bacterial attachment, to determine if biocompatibility and antibacterial properties can be improved upon simultaneously using fibre laser treatment.

## 2. Experimental Section

### 2.1. Materials

Ti-35Nb-7Zr-6Ta plates were sourced (American Element, Los Angeles, CA, USA) with dimensions of 250 mm × 250 mm with 3 mm thickness, and were wire cut using electrical discharge machining (EDM) (Kaga, Ishikawa, Japan) into 30 mm × 40 mm plates. The material was polished prior to laser treatment using a progression of silicon carbide (SiC) papers with a finish of 1000 grit. Standard metallographic procedures were followed to remove the pre-existing oxide layer and any surface defects present after the manufacturing process. Samples were ultrasonically cleaned in acetone for 10 min, rinsed with deionised water and air-dried prior to laser treatment and material characterisation. A sample size of n = 3 was used for all material characterisation and in vitro cell culture experiments except bacterial attachment, for which n = 4 samples were used.

### 2.2. Laser Treatment

Laser surface treatment was performed using an automated continuous wave (CW) 200 W fibre laser system (MLS-4030). The laser system was integrated by Micro Lasersystems BV (Driel, Gelderland, the Netherlands) and the fibre laser was manufactured by SPI Lasers UK Ltd (Southhampton, Hampshire, UK). The laser wavelength was 1064 nm. The samples were prepared using the following parameters: laser power 30 W, stand-off distance 1.5 mm, argon gas with 30 L/min flow rate and two different scanning speeds, 100 and 200 mm/s. Laser sample groups are denoted as LT100 and LT200 hereafter. The laser-treated area of the surface was 6 mm² in square shape. The laser energy at the two speeds was 1.8 and 0.9 J respectively (see Supplementary Materials for calculations). The control base metal samples (1000 grit finish) are denoted as BM. The sample plate was used for bacterial attachment, otherwise samples were wire cut using EDM into 6 mm diameter discs. Prior to biological culture, samples were ultrasonically cleaned in acetone twice for 1 h, then in deionised water for 30 min and air-dried in the fume hood before a final sterilisation step in an autoclave (Prestige Medical, Blackburn, Lancashire, UK) at 121 °C and 1.5 bar pressure for 20 min to destroy any microorganisms present on the surface.

### 2.3. Surface Roughness, Topography and Composition

The surface roughness and 3D profile of the untreated and laser treated samples were captured using white light interferometry (WLI) (Talysurf CCI 6000, Leicester, Leicestershire, UK). Roughness was assessed using four parameters: arithmetic mean (Ra), maximum profile height (Rz), surface skewness (Rsk) and surface kurtosis (Rku). Values were extracted from the 1.2 mm$^2$ scan areas perpendicular to the laser track orientation. Scanning electron microscopy (SEM) was used to image the ripples on the laser treated surfaces (FlexSEM 1000, Hitachi, Maidenhead, Berkshire, UK). SEM images were acquired using a 20 kV beam and backscattered electron compositional (BSE-COMP) mode detection. SEM images for energy dispersive X-ray spectroscopy (EDX) analyses were acquired using a Zeiss Leo 1455VP SEM at 20 kV beam energy with secondary electron detection. EDX data were acquired in the SEM using an Oxford Instruments X-Act detector (Abingdon, UK) with INCA v4.15 acquisition and processing software.

### 2.4. Phase Identification

Phase and crystallographic structure of the samples were captured by X-ray diffraction (XRD) using a PANanalytical X'Pert Pro MPD (PANalytical, Tollerton, Nottingham, UK) with a CuK$\alpha$ radiation source operated at 40 kV, 40 mA with $\frac{1}{2}°$ fixed slit, 10° anti-scatter slit and 0.02 step size with Ni filter. Samples were analysed in a 2 theta (2θ) range between 10°–90°.

### 2.5. Surface Chemistry

X-ray photoelectron spectroscopy (XPS) spectra were acquired using a bespoke ultra-high vacuum (UHV) chamber fitted with Specs GmbH Focus 500 monochromated Al K$\alpha$ X-ray source and Specs GmbH Phoibos 150 mm mean radius hemispherical analyser with 9-channeltron detection. Survey spectra were acquired over the binding energy range between 0 and 1100 eV using a pass energy of 50 eV, and the high resolution scans over the C 1s, Ti 2p, Zr 3d, Nb 3d and O 1s lines were made using a pass energy of 20 eV. Data were quantified using Scofield cross-sections corrected for the energy dependencies of the effective electron attenuation lengths and the analyser transmission. Data processing and curve fitting were carried out using the CasaXPS software v2.3.16 (CasaXPS, Teignmouth, Devon, UK).

### 2.6. In Vitro Cell Culture

#### 2.6.1. Attachment

Cell culture was performed in a Class II microbiological safety cabinet, and sterile conditions were maintained. Human mesenchymal stem cells (passage 5–9) (Texas A&M Health Science Centre College of Medicine, Institute for Regenerative Medicine, Bryan, TX, USA) were cultured in tissue culture flasks (Thermo Scientific). The medium was comprised of Minimum Essential Medium Alpha with GlutaMAX (Gibco), supplemented with 16.5% foetal bovine serum and 1% glutamine. The cells were maintained in a humidified atmosphere with 5% CO$_2$ at 37 °C, and were sub-cultured when they reached confluency by washing with phosphate buffered saline (PBS) and disassociated with 0.05% Trypsin-EDTA (Gibco) to provide adequate cell numbers for all studies undertaken, or for further subculture or cryopreservation. Cells were counted using a haemocytometer (Agar Scientific), and seeded in a 96 well plate (Sarstedt) at a density of $5 \times 10^3$ cells per well and maintained in the same culture conditions as previously mentioned. Early cell attachment was assessed at the following times: 6, 24 and 48 h using direct immunofluorescent staining, the procedure for which is as follows. Cells were quickly washed with cold PBS, fixed with 4% paraformaldehyde (PFA) for 15 min and permeabilised with 0.1% Triton X-100 in PBS for 5 min. Cells were then blocked with 5% donkey serum in PBS for 30 min. The cells were stained with α smooth muscle actin (SMA)-Cy3 conjugated mouse antibody (1:200) in 5% donkey serum in PBS for 45 minutes at 37 °C. Cells were then counterstained with 4′,6-diamidino-2′-phenylindole dihydrochloride DAPI (1:1000) in PBS for 5 min.

Each step was performed at room temperature unless otherwise stated. Cells were washed with PBS in between every step except after blocking. The last step involved a final wash with distilled water. Samples were transferred to a new well plate for imaging using the Leica DMi8 inverted fluorescence microscope (Leica, Wetzler, Hesse, Germany). Three images were captured per sample at magnification ×100, giving a total of nine images per group at each time point.

### 2.6.2. Proliferation

Cell proliferation capacity was assessed using the CyQUANT NF Cell Proliferation Assay Kit (Thermo Scientific) at 3, 7 and 14 days. The dye solution was prepared by adding 1× dye binding solution to 1× Hank's balanced salt solution (HBSS). The cell culture medium was aspirated and 200 μl of the dye solution was added to each well and incubated at 37 °C for 30 min. The fluorescence intensity was measured using a microplate reader (Varioskan LUX) at 485 nm excitation and 530 nm emission.

### 2.6.3. Differentiation

Cells were stained using indirect immunofluorescent assay staining at 7, 14 and 21 days. All steps were the same as previously mentioned, see Section 2.6.1, except after blocking, where cells were instead incubated with anti-osteocalcin (10 μg/ml) (R&D Systems) in 5% donkey serum in PBS for 1 h. Cells were then stained with Alexa Fluor 488 donkey anti-mouse (1:500) (Thermo Scientific) in PBS for 45 min. This was followed by DAPI staining, PBS wash and distilled water wash. Samples were transferred to a new well plate for imaging using the Leica DMi8 inverted fluorescence microscope (Leica, Wetzler, Hesse, Germany). Three images were captured per sample at magnification ×100, giving a total of nine images per group at each time point.

### *2.7. Bacterial Attachment*

The TNZT plate was washed three times with sterile PBS. *S. aureus* (ATC 44023) was cultured in Müller Hinton broth (MHB) for 18 h at 37 °C on a gyrotatory incubator with shaking at 100 rpm. After incubation, sterile MHB was used to adjust the culture to an optical density of 0.3 at 550 nm, and it was diluted (1:50) with fresh sterile MHB. This provided a bacterial inoculum of approximately $1 \times 10^6$ colony forming units (CFU)/mL. 1 mL of culture was applied to the plate carefully suspended over a petri dish base at an inoculum not exceeding $2.4 \times 10^6$ CFU/mL, as verified by viable count. The plate was incubated for 24 h at 37 °C on a gyrotatory incubator with shaking at 100 rpm. Four samples of each group, for both untreated and laser treated, were tested to ensure the consistency of the results. After 24 h of incubation, the plate was washed three times with sterile PBS to remove any non-adherent bacteria. The adherent bacteria were stained with fluorescent Live/Dead®BacLight™ solution (Molecular Probes) for 30 min at 37 °C in the dark. The fluorescent viability kit contains two components: SYTO 9 dye and propidium iodide. The SYTO 9 labels all bacteria, whereas propidium iodide enters only bacteria with damaged membranes. Green fluorescence indicates viable bacteria with intact cell membranes, while red fluorescence indicates dead bacteria with damaged membranes. The stained bacteria were observed using a fluorescence microscope (GXM-L3201 LED, GX Optical, Stansfield, Suffolk, UK). Twenty four random fields of view (FOV) were captured per group. The surface areas covered by the adherent bacteria were calculated using ImageJ software. The areas corresponding to the viable bacteria (coloured green) and the dead bacteria (coloured red) were individually calculated. The total biofilm area was the sum of the green and red areas, and the live/dead cell ratio was the ratio between the green and red areas. The results are expressed as the means of the twenty four measurements taken per group.

### *2.8. Statistical Analysis*

Data were expressed as mean ± standard error (SE). The significance of the observed difference between untreated and laser treated samples was tested by one-way and two-way ANOVA using Prism software (GraphPad Prism Software Version 7, San Diego, CA, USA). Statistically significant

differences were calculated, with a *p*-value of ≤0.05 considered significant; (*) $p < 0.05$; (**) $p < 0.01$; (***) $p < 0.001$.

## 3. Results

### 3.1. Surface Roughness by WLI

The 3D and 2D profiles of the untreated and laser treated surfaces can be seen in Figure 1A,B. The laser treated surfaces appear to have very smooth tracks (as indicated by each individual colour scale), showing a higher degree of surface polishing effect with increased laser scanning speed, as shown by the shift in the colour scale of the 2D profiles. It is also evident that the base metal areas in between the laser tracks have significant variation in surface roughness, as is expected of the metallographic preparation process (i.e., polished by SiC paper). The roughest regions are prominent at the very edge of the laser tracks for each surface (as indicated by the arrows in Figure 1A,B).

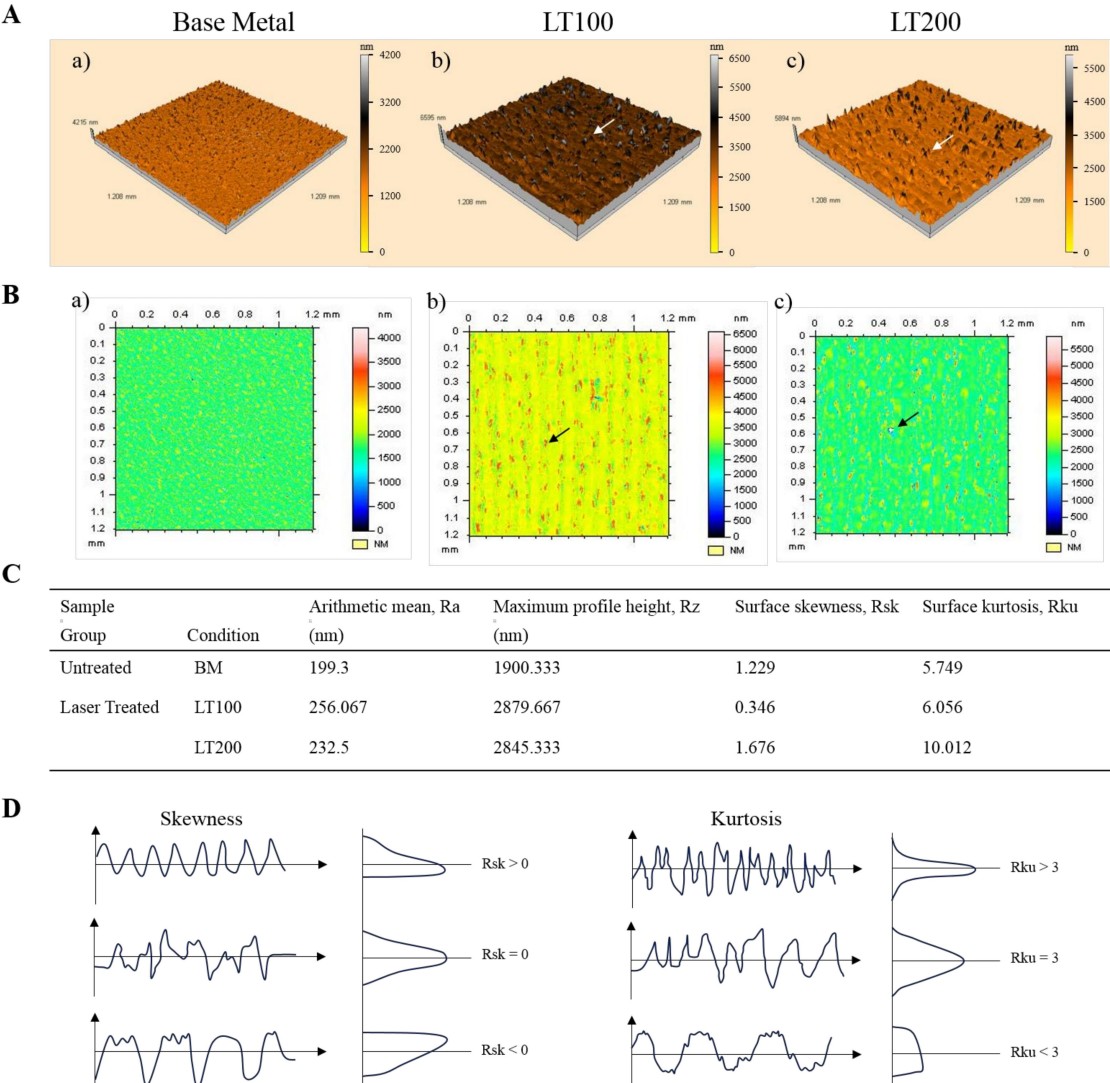

**Figure 1.** Laser surface treatment has a polishing effect, which increases with laser scanning speed while creating a rough and spiky laser track edge. White light interferometry (**A**) 3D and (**B**) 2D profiles for the untreated and laser treated samples. Scan area of 1.208 mm² and 1.2 mm² for 3D and 2D images respectively; colour coding of each image seen by individual colour scale in figure. Arrows show the spiky areas at the laser track edges. (**C**) Summary of surface roughness parameters. (**D**) Diagrams illustrating visual difference between surface skewness and kurtosis profiles.

The surface roughness values extracted from the 2D profiles can be seen in Figure 1C. The Ra values ranged from a low of 199.3 nm on BM to a highest value of 256.1 nm on the LT100 surface, while LT200 had a Ra of 232.5 nm. The Rz values followed a similar trend to the Ra, with BM having the lowest maximum profile height (1900.3 nm) while LT100 and LT200 had similar values, 2879.7 and 2845.3 nm, respectively. Overall, the laser surface treatment increased the Ra and Rz in comparison to the untreated surface, due to the rough edges created on the edges of the laser created tracks.

The untreated and laser treated samples were also quantified using the surface skewness and kurtosis values. The skewness defines whether a surface consists of spikes (Rsk > 0) or valleys (Rsk < 0), while the kurtosis defines whether a surface has peaks (Rku > 3) or is flat (Rku < 3); see Figure 1D for diagram illustrating the visual differences between varying Rsk and Rku values. There was a slight increasing trend in Rsk and a more notable difference in Rku as the laser scan speed increased, as seen in Figure 1C. The LT100 surface had the lowest Rsk (0.346), while the BM had a value of 1.229. The LT200 had the highest Rsk with 1.676, showing that the highest scanning speed creates the spikiest surface. All Rku values were >3, with BM being the lowest at 5.749 and increasing with scanning speed. There was a sharp increase in Rku at the highest scanning speed, reinforcing the laser surface polishing effect. All the surfaces have a Rsk > 0 and Rku > 3. The LT200 group had the highest skewness and kurtosis, suggesting it had more peaks and spikes present on the edges of the laser created tracks, but the smoothest laser treated area.

## 3.2. Surface Topography and Composition by SEM-EDX

The SEM images of the untreated and laser treated surfaces can be seen in Figure 2A(a–d). The typical surface morphology after mechanical grinding can be seen above and below each laser track in Figure 2A(a,b), with the surface exhibiting random scratches, pits and grooves. The tracks created using laser treatment had a distinctive ripple effect, which was more prominent at the lower scan speeds, with small ripples present along the entire track and the periodic appearance of distinctive larger ripples. The magnified SEM images show that the ripples were much smaller and uniform on the LT200 surface than on LT100, which had more distinct arches, as shown in Figure 2A(c,d). The tracks created by laser surface treatment became smoother as the scanning speed increased, verifying the laser surface polishing effect observed in the WLI images.

The ripple effect was quantified by calculating the peak-to-peak distance between ripples, using Image J, as seen in Figure 2B,C. The distance became smaller with increasing laser scanning speeds. The peak-to-peak distance between ripples halved when comparing LT100 and LT200, dropping from 0.63 to 0.315 μm. The surfaces were significantly different from one another ($p < 0.001$).

The SEM-EDX analyses were performed to determine if there were any notable differences in elemental distribution post laser treatment, and can be seen in Figure 2D. The base metal polished areas were quite distinctive from the smooth laser treated tracks; the boundary between these areas is defined by a dashed yellow line. The carbon and oxygen were quite densely concentrated on the base metal polished area of the samples, while homogeneously distributed in the laser tracks on each surface, irrespective of scanning speed. It is apparent that for each element, except Ta, the laser treatment created a homogenous elemental distribution within the tracks with no measureable spatial variation. Enriched particles of Ti, Nb and Zr can be seen in Figure 2D(i–vi) for each surface, as indicted by the black arrows, which can be attributed to the base metal polished surface features. There was a more uniform and homogeneous oxide film present post laser treatment.

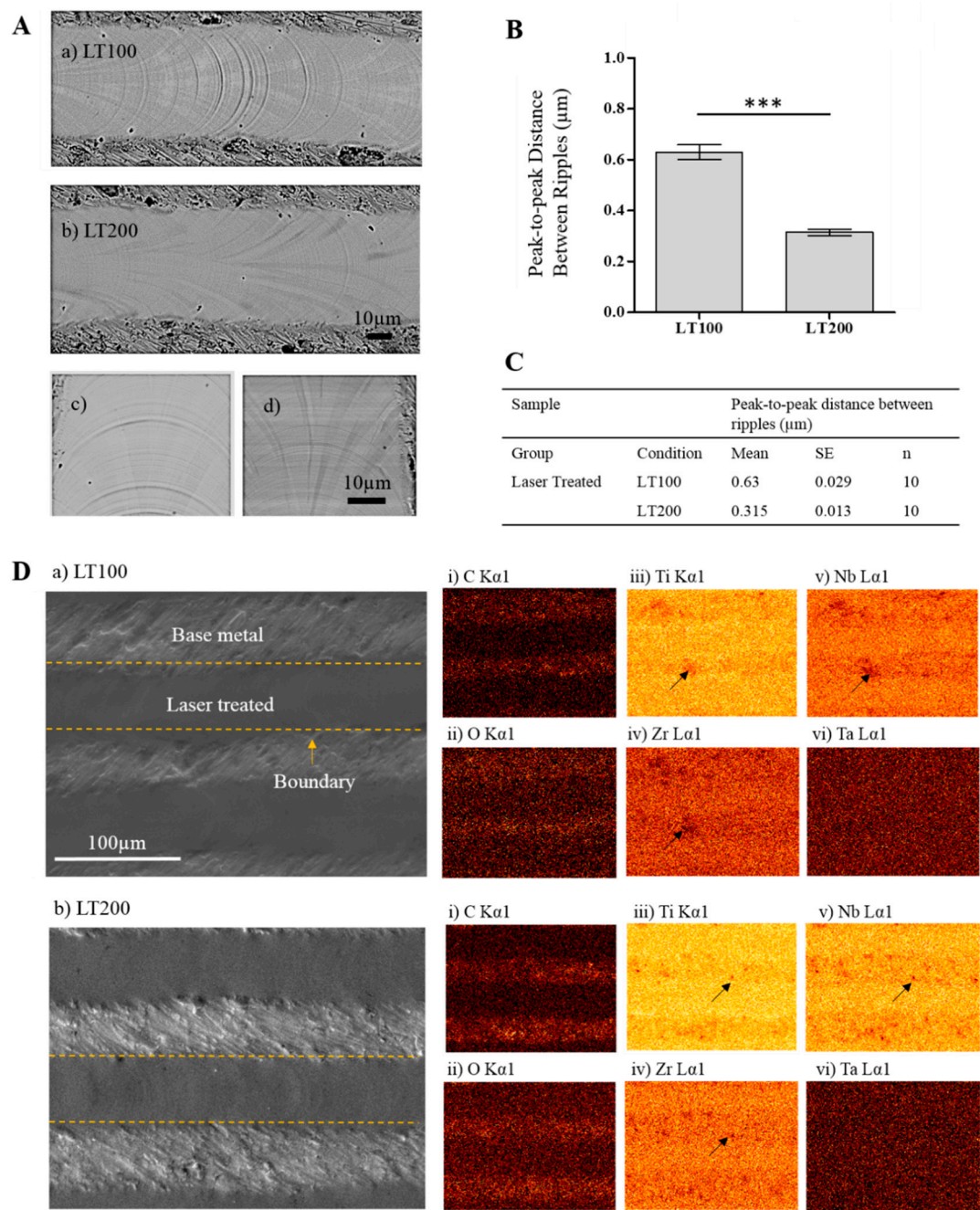

**Figure 2.** Distinctive microscale ripples and homogenous elemental distribution achieved through laser surface treatment. (**A**) SEM images of the multi-scale ripples created within the laser tracks. Scale bar = 10 μm. (**B**) Peak-to-peak distance between ripples, calculated using ImageJ. Error bars are the SE of n = 10. (**C**) Tabulated peak-to-peak distance between ripples. (**D**) SEM-EDX images showing the laser treated surfaces, with a boundary defining the base metal polished and laser treated areas. Scale bar = 100 μm. EDX elemental maps show the spatial distribution of carbon, oxygen, titanium, niobium, zirconium and tantalum. Black arrows indicate particle enrichment in the base metal polished zones for all elements except tantalum.

### 3.3. Phase Identification by XRD

The phase and structure of the samples were identified using XRD, as seen in Figure 3. There was a notable preferential crystallographic phase shift post laser treatment. The blue dot indicates the presence of a specific peak in all samples. The untreated base metal surface had a prominent

peak at ~56° β (200), with additional smaller peaks present (~38.5° β (110) and ~69° β (211)). The crystallographic plane shifted to ~38.5° β (110) after laser treatment for LT100. All samples had a peak present at ~38.5° β (110), ~56° β (200) and ~69° β (211), although LT100 had an additional weak peak present at ~83° (β 220). All peaks were associated with beta phases, no alpha phases were detected. This is due to the presence of beta stabilising elements (niobium and zirconium) in the material, which suppresses the formation of alpha phase.

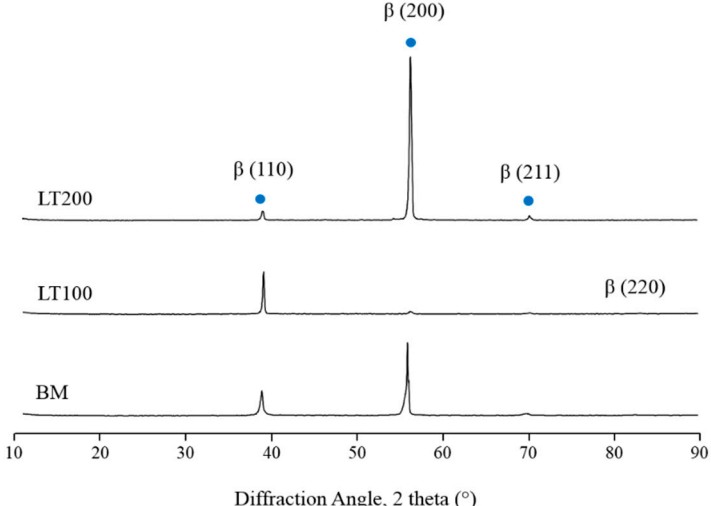

**Figure 3.** Shifting crystallographic plane with laser surface treatment. XRD profiles of the untreated and laser treated samples. The scanning diffraction angles are between 10° and 90°. Y-axis shows intensity (a.u.).

*3.4. Surface Chemistry by XPS*

The XPS spectra and narrow scans for the untreated and laser treated groups can be seen in Figure 4. A summary of the % concentrations by XPS on untreated and laser treated surfaces can be found in Table 1, and a summary of the species detected are noted in Table 2. All surfaces showed the expected Ti, Nb, Zr, C and O, with the additional presence of N, Zn and Cu. The presence of organic nitrogen is probably due to sample exposure to air [32]. The Cu and Zn on the BM surface were attributed to surface contamination during manufacture of the alloy, and showed levels which reduced as a consequence of laser treatment. Each element and assignment, as seen in Table 2, was found in all of the laser treated samples.

**Table 1.** Summary of % concentrations by XPS on untreated and laser treated surfaces.

| | XPS (% Concentration) | | |
|---|---|---|---|
| **Name** | **BM** | **LT100** | **LT200** |
| C 1s | 54.1 | 56.4 | 52.2 |
| N 1s | 7.7 | 4.4 | 3.2 |
| O 1s | 27.4 | 30.8 | 32.9 |
| Si 2p | 1.8 | 0.5 | 0.6 |
| Ti 2p | 3.4 | 6.1 | 8.6 |
| Cu 2p3/2 | 2.7 | 0.2 | 0.8 |
| Zn 2p3/2 | 0.8 | 0.6 | 0.4 |
| Zr 3d | 0.4 | 0.5 | 0.7 |

The untreated sample had no Ti $2p_{3/2}$ in the form of Ti metal present in the surface layer, and a very weak presence of $Ti^{3+}$ in $Ti_2O_3$/TiN at 456.2 eV. The majority of Ti at the outermost untreated surface was found in the $Ti^{4+}$ state in $TiO_2$ at 458.4 eV. The case was similar for the two other metal elements,

niobium and zirconium. Nb was only found in the $Nb^{5+}$ state in $Nb_2O_5$ at 207 eV, while the majority of Zr was found in the $Zr^{4+}$ assignment in $ZrO_2$ at 182.3 eV. There was very little to no nitride present on the untreated sample, whereas there was some evidence from the laser treated samples to suggest that the laser treatment creates a nitride layer at a BE range of 395.8–397 eV.

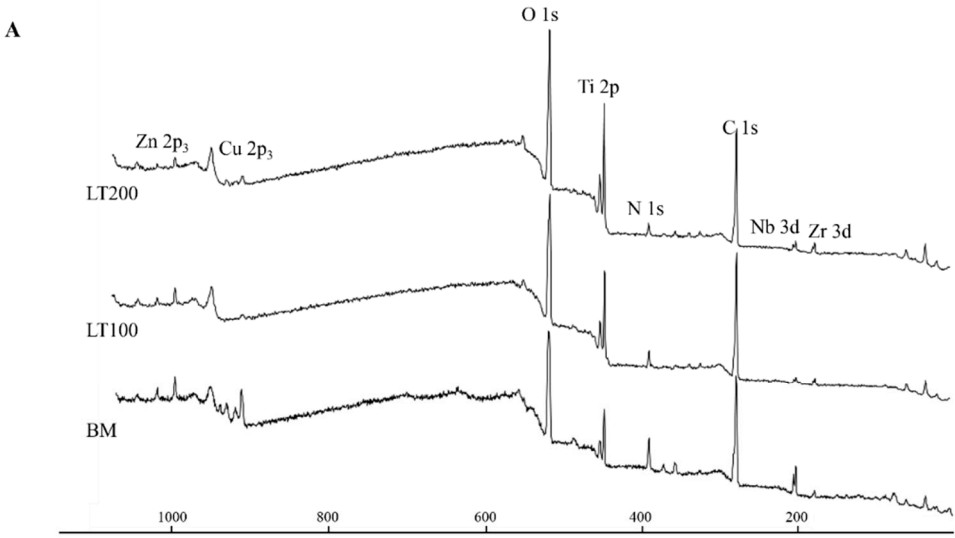

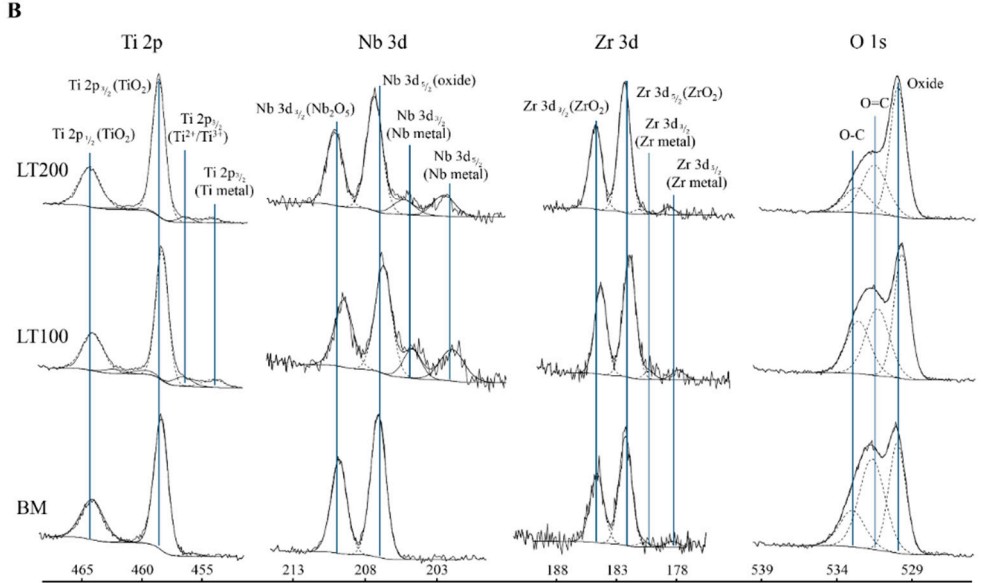

**Figure 4.** XPS spectra of untreated and laser treated samples. (**A**) XPS spectra: Zn 2p₃, Cu2p₃, O 1s, Ti 2p, N 1s, C 1s, Nb 3d and Zr 3d. (**B**) XPS narrow scan spectra: Ti 2p, Nb 3d, Zr 3d and O 1s for untreated and laser treated surfaces. X-axes show binding energy (eV).

The oxygen spectra for the untreated and laser treated samples typically showed a relatively sharp lower binding energy metal-bonded component, and a broader but relatively featureless higher binding energy component typical of organic oxygen, e.g., bonded within the hydrocarbon contamination layer. The metal-bonded component (i.e., Ti–O, Nb–O and Zr–O) was typically found at 529.8 eV. The carbon–oxygen region was typically fitted with two components at 530.9–531.5 eV and 532.3–532.9 eV, representative of O=C and O–C bonding respectively, typically associated with residual organic contamination.



**Table 2.** Summary of the chemical species detected by XPS.

| Element | Line | Assignment | BE Range (eV) | Present in Sample | | |
|---------|------|------------|---------------|------|-------|-------|
|         |      |            |               | **BM** | **LT100** | **LT200** |
| Ti | $2p_{3/2}$ | Ti metal | 454.3 | No | Yes | Yes |
|    |            | $Ti^{3+}$ in $Ti_2O_3/TiN$ | 456.2 | V. weak | Yes | Yes |
|    |            | $Ti^{4+}$ in $TiO_2$ | 458.4 | Yes | Yes | Yes |
| O | 1s | Ti–O, Nb–O, Zr–O | 529.8 | Yes | Yes | Yes |
|   |    | O=C | 530.9–531.5 | Yes | Yes | Yes |
|   |    | O–C | 532.3–532.9 | Yes | Yes | Yes |
| C | 1s | C–C | 285 | Yes | Yes | Yes |
|   |    | C–O | 286–286.4 | Yes | Yes | Yes |
|   |    | C=O | 287.2–288.1 | Yes | Yes | Yes |
|   |    | COO- | 288.6–289.3 | Yes | Yes | Yes |
| Nb | $3d_{5/2}$ | Nb metal | 202–202.5 | No | Yes | Yes |
|    |            | $Nb^{5+}$ in $Nb_2O_5$ | 207 | Yes | Yes | Yes |
| Zr | $3d_{5/2}$ | Zr metal | 178.4 | Weak | Yes | Yes |
|    |            | $Zr^{4+}$ in $ZrO_2$ | 182.3 | Yes | Yes | Yes |
| N | 1s | Nitride | 395.8–397.0 | V. weak/No | Yes | Yes |
|   |    | Organic | 399.7–400.3 | Yes | Yes | Yes |

## 3.5. Cell Responses

### 3.5.1. Attachment

It is evident at the early attachment time points that the cells behaved distinctively differently on the laser treated surfaces in comparison with the untreated surfaces between the 24 and 48 h time points, as seen in Figure 5. Alpha SMA was specifically used to clearly visualise cell morphology. The MSCs were visually similar in shape at 6 h on all surfaces, displaying the typical polygonal structure with uniform spreading regardless of underlying surface topography. At 24 and 48 h, the cells on the laser treated surfaces began to show evidence of interacting with the surface. At 24 h, the LT100 cells remained fairly rounded and polygonal in shape, while the LT200 surfaces encouraged cell stretching, seen across and along the laser created tracks. At 48 h, the cell shapes had vastly changed again, with LT100 causing the cells to become slightly smaller in size in comparison with their untreated BM counterparts. LT200 appears to have the most influence on cell shape, as it is clear that at 48 h the cells displayed a spindle shaped appearance. Meanwhile, the BM surface encourages cells to stretch. This is probably an effect of the scratch marks remaining after the SiC paper polishing process.

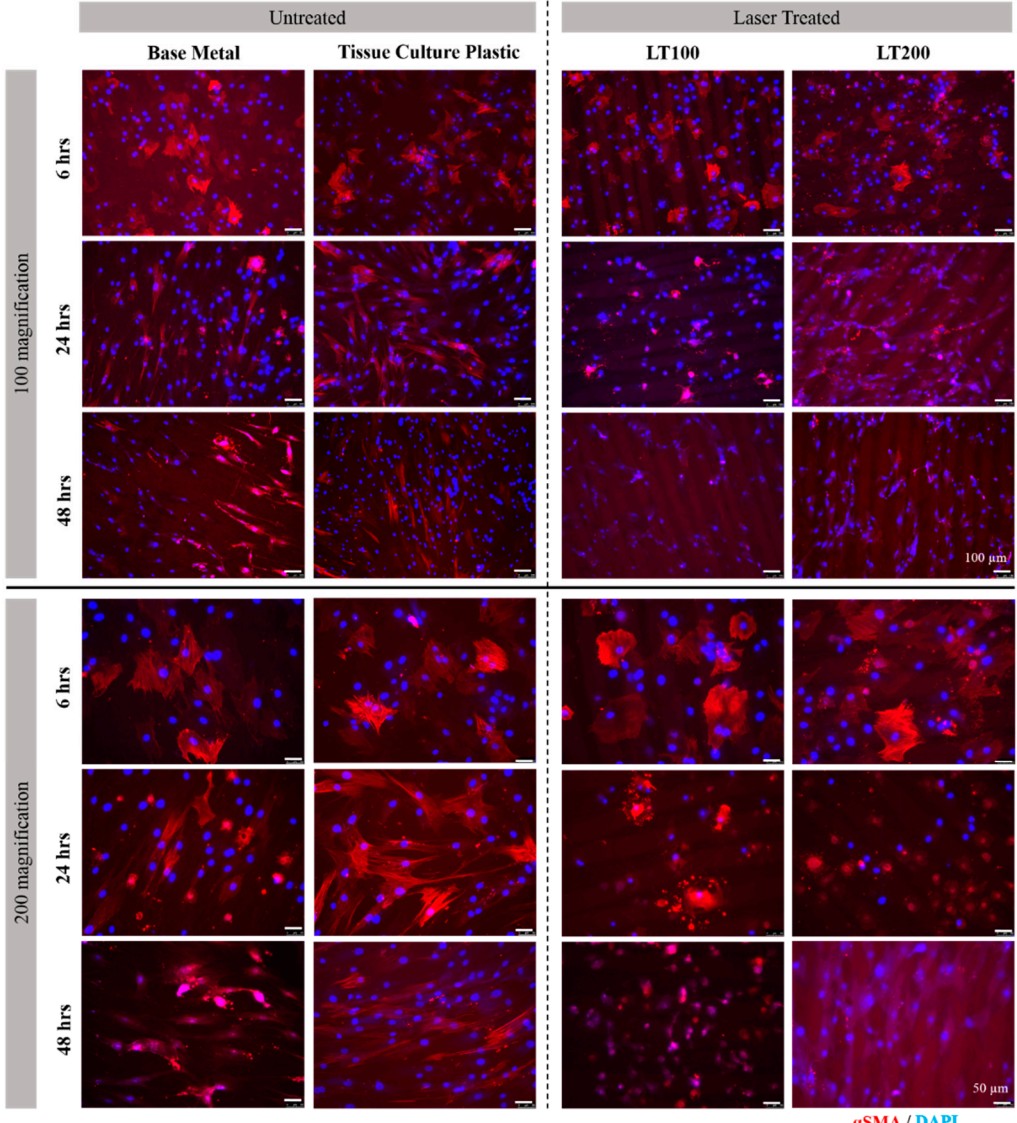

**Figure 5.** Laser surface treatment at high scanning speeds can be used to encourage cell spreading. Merged fluorescence images of mesenchymal stem cells (MSCs) stained at 6, 24 and 48 h at 100× (top) and 200× (bottom) magnification on untreated and laser treated surfaces. Red = αSMA actin fibres; blue = DAPI nuclei stain. Scale bar = 100 μm (top) and 50 μm (bottom). Images are representative of cell coverage on the entire surface.

### 3.5.2. Proliferation

The fluorescence intensity of MSCs on the untreated and laser treated surfaces at Day 3, 7 and 14 can be seen in Figure 6. A higher intensity is associated with more cells being present on the surface. There were no significant differences between surfaces within time points until Day 7 and 14. At Day 7, there was a significant difference between BM and LT100 ($p < 0.05$). At Day 14, the untreated BM had the highest fluorescence intensity, followed by LT200 then LT100. BM was significantly different from the two laser treated groups, LT100 ($p < 0.001$) and LT200 ($p < 0.05$).

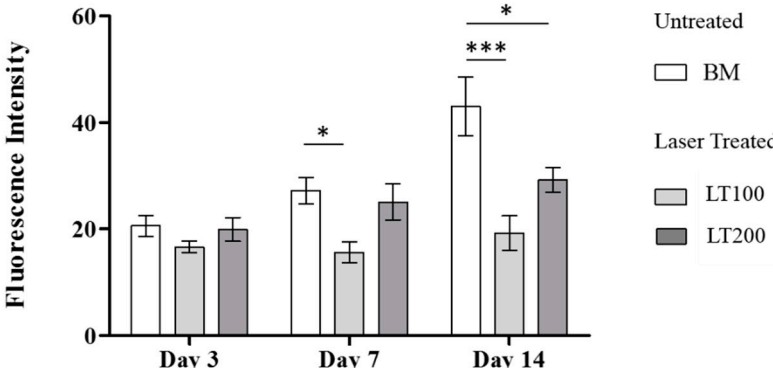

**Figure 6.** Fluorescence intensity of MSCs at Day 3, 7 and 14 proliferation time points on untreated and laser treated surfaces. The error bars indicate the SE of n = 3.

### 3.5.3. Differentiation

MSC morphology was qualitatively assessed using fluorescence staining of osteocalcin at Day 7, 14 and 21. There was a very distinct difference in MSC morphology on the untreated and laser treated surfaces, as seen in Figure 7. Those on BM were spindle shaped, while TCP had good coverage, as expected. More distinct cell shapes can be seen on LT200 than on LT100. The distinctly different cell morphology can be seen at the later differentiation time points. The number of cells increased on all surfaces between Day 7 and Day 14, with TCP having nearly full coverage and monolayer. The cells on BM had the same spindle shape, while the cells on both laser treated surfaces had begun to form clusters, the larger seen on LT200, with small rounded cells present alongside the clusters.

At Day 21, the TCP had formed a monolayer of MSCs, while the cell clusters could be seen on the BM surface. There were multiple cell cluster formations on the laser treated surfaces, which could be indicative of bone-like nodule formation, suggesting that the parameters chosen for the laser surface treatment encourage bone to grow more quickly than on the BM surface, suggesting that laser surface treatment is a potential modification technique to encourage faster bone growth.

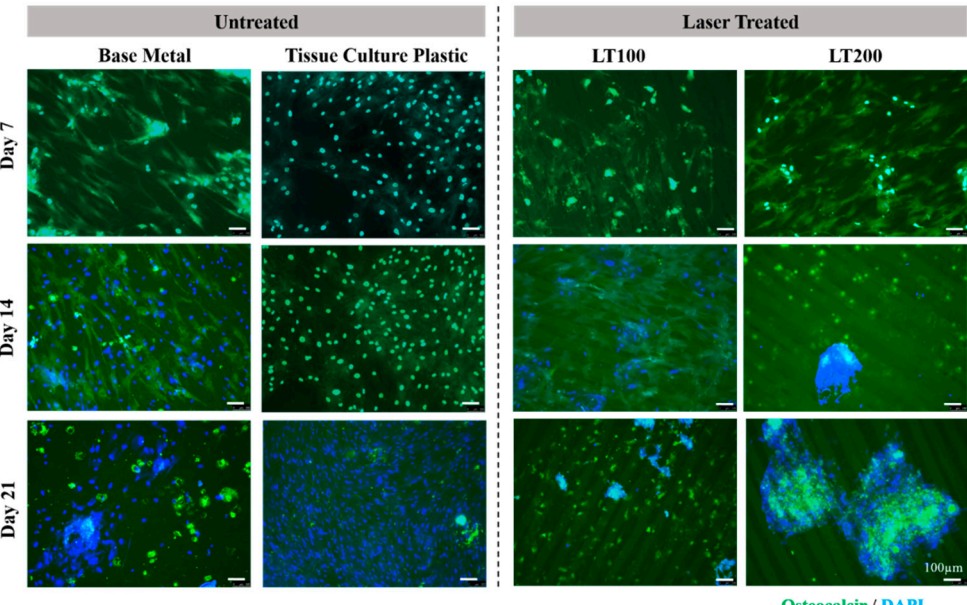

**Figure 7.** Laser surface treatment encourages formation of bone-like nodules, indicating potential faster osseointegration. Merged fluorescence images of MSCs stained at Day 7, 14 and 21 at 100× magnification on untreated and laser treated surfaces. Green = osteocalcin protein; blue = DAPI nuclei stain. Scale bar = 100 μm. Images are representative of cell coverage on the entire surface.

### 3.6. Bacterial Attachment

The bacterial attachment results on the untreated and laser treated surfaces can be seen in Figure 8. Attachment was quantitatively analysed using *S. aureus* coverage and live-to-dead ratio to determine which surface(s) elicited a bactericidal response. The fluorescence images show that live cells were green stained with SYTO 9, while dead cells were red stained with propidium iodide. Green fluorescence indicates viable bacteria with intact cell membranes, while red fluorescence indicates dead bacteria with damaged membranes. As seen in Figure 8A, there was a visibly higher number of green stained cells present on the untreated surface. After only 24 h, the lower number of bacteria on the laser treated surface in comparison with the untreated suggests that laser treatment created an inhospitable environment for the bacteria, causing them to become non-viable. The bacterial attachment results were quantified using bacteria coverage and the ratio of live-to-dead cells. The bacterial coverage on each laser treated surface was significantly different ($p < 0.001$) from the untreated BM surface, see Figure 8B. There was a four-fold increase in the bacterial coverage between the laser treated and untreated group. Likewise, the live/dead ratio of bacteria present on the laser treated surfaces was also drastically reduced, see Figure 8C.

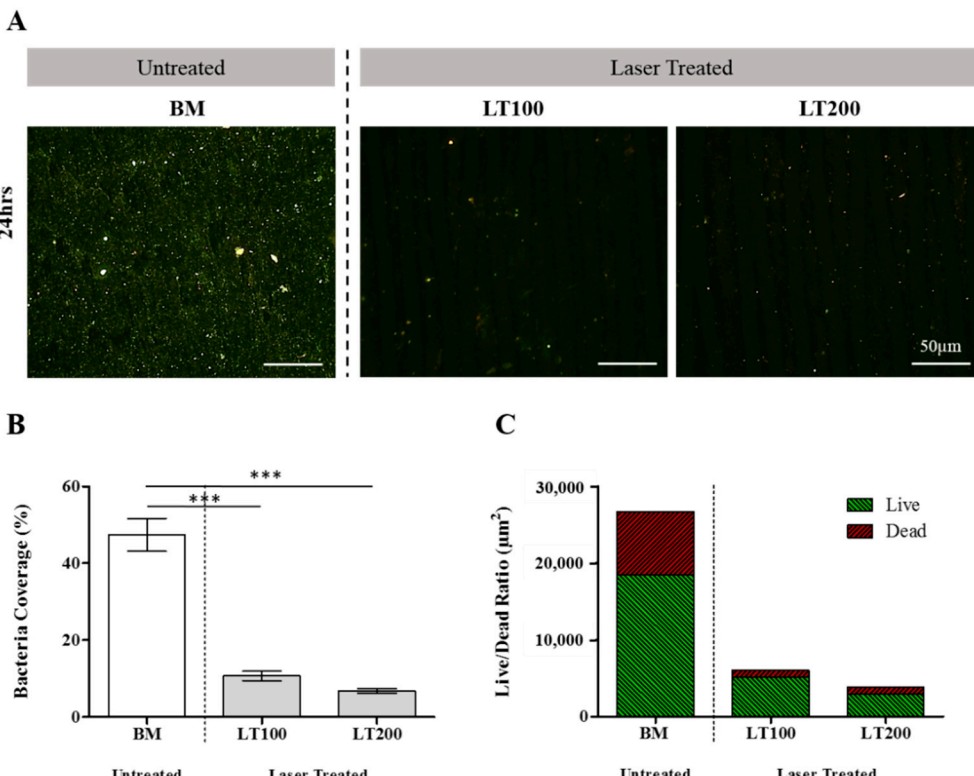

**Figure 8.** Bacterial adherence significantly reduced with laser surface treatment. (**A**) Merged fluorescence images of bacteria at 24 h using live/dead staining. Scale bar = 50 μm. Images are representative of bacteria coverage on the entire surface. (**B**) Total bacteria coverage measured from fluorescence microscopy. Error bars are the SE of n = 24. The total bacteria coverage refers to the total sum of red and green areas present after cell staining. (**C**) Live-to-dead ratio of bacteria cells measured from fluorescence microscopy on the surfaces post staining, based on average red and green areas per group. Data calculated in both graphs using twenty four images per surface at each time point using ImageJ software.

## 4. Discussion

Bimodal texturing is an important consideration for improvement in surface feature design. Micro and nanotopography must be used in tandem to create a surface that is sufficient for long-term stability [19,33]. Micro scale features such as grooves, ridges and pits can increase surface area

and provide more opportunities for attachment, and these features can cause cells to align and organise within them. Features at the nano scale directly affect protein interactions, filaments and tubules, which control cell signalling and regulates cell adhesion, proliferation and differentiation [34].

The roughest regions were prominent at the very edge of the laser tracks for each surface (as indicated by the arrows in Figure 1A,B). This can be attributed to the consequence of melt pool dynamics at the liquid/solid boundary (or melt pool/heat affected zone (HAF) boundary) during the laser re-melting process [35]. It is known that the micro-ripple surface (as seen in Figure 2A) results from oscillation of the liquid metal due to the Marangoni convection and hydrodynamic processes driven by thermocapillary motion acting on the melt pools [36,37]. The black marks seen in Figure 2A(a–d) could be due to contamination from the material handling process, although further in-depth analysis is required.

There was a more uniform and homogeneous oxide film present post laser treatment (Figure 2D). The elements on the LT100 and LT200 surfaces were uniformly distributed with no measureable spatial variation over the surface area imaged, suggesting the elongated and thin brush-like marks (extended from the interior areas of laser tracks to the boundary, as seen in Figure 2A) were laser-induced surface features as a consequence of the convection field in the complex melt pool dynamics and solidification processes during laser treatment [38]. Tantalum is the only element that did not show any evidence of particle enrichment, perhaps due to it being the least abundant element present in the quaternary alloy.

There was a notable preferential crystallographic phase shift post laser treatment (Figure 3). This could be due to the preferential orientation of a specific phase (i.e., peak angle of ~38.5°) caused by higher laser energy input at lower scanning speed. However, further in-depth analysis is still required to investigate how the difference of laser energy input between the two scanning speeds (i.e., lower at 100 mm/s and higher at 200 mm/s) makes the change. Sharp dominant phase peaks present in XRD analysis indicated the treated material had a high degree of crystallinity.

The significant reduction of bacterial attachment and/or biofilm formation on the laser treated surfaces can be attributed to the following: (i) Firstly, the SEM-EDX results indicated "homogenisation" or "finer dispersion" of metal compositions in the laser-melted surfaces (i.e., absence of metal-enriched particles, as indicated in Figure 2D). In addition, the laser treated surfaces look smoother in the SEM images (Figure 2A), and the original fine-scale texture and roughness of the untreated areas was not present after laser treatment. Both of these can be linked to the formation of a more uniform and homogeneous oxide film after laser treatment. (ii) Secondly, the XPS data and also the SEM-EDX results indicate that overall the oxide film was thinner after laser treatment, but it was likely to be more uniform in thickness, as described in point (i) above. Therefore, possibly, there was a better coverage of a more well-defined though thinner oxide layer over the treated areas. (iii) Thirdly, removal of residual organic contaminants, as indicated by the reduced proportion of both oxygen-bonded and carbon-bonded species in the oxide film, which could be acting as potential sources to attract bacteria to attach on surfaces via non-covalent interactions [39]. The phenomenon of laser treatment helps reduce overall levels of organic surface contaminants, as reported elsewhere [16,36]. This can be due to rapid vaporisation of more volatile species under the sudden input of energy from the laser, or, possibly, more adherent contaminants are buried as the locally-melted metal re-solidifies after laser treatment. (iv) Finally, the presence of titanium nitride (TiN) in the oxide film after laser treatment could contribute to the antibacterial activities. It has been reported that a surface with TiN can deactivate biofilm formation [14,40]. Likewise, zirconium nitride (ZrN) is known to be an antibacterial material [41]. However, the possibility of an antibacterial effect attributed to ZrN can be eliminated in this study, because there is no evidence for the existence of ZrN in the oxide film, as shown in the XPS narrow scan profile in Figure 4B (i.e., the binding energy for ZrN would be expected to be around 180.9 eV [42]). In contrast, the evidence for TiN present in the oxide film after laser treatment is clear (i.e., the curve fitted at the binding energy around 456.2 eV in Figure 4B).

It is important to note that, although the evidence for the appearance of metallic species in the oxide film, namely Ti and Nb metals, after laser treatment is also very clear, the results in this study indicated that they are not necessarily encouraging the bacterial attachment and/or biofilm formation. The presence of the oxide layer improves the corrosion resistance of the material's passive surface [43]. The organic contamination found in the form of O–C and O=C bonds could be due to the material handling process or a carbon-containing cleaning agent; further in-depth analysis is required.

The ultrastructure of the bone–titanium interface demonstrates simultaneous direct bone contact, osteogenesis and bone resorption [44]. Upon implantation, a material surface initially interacts with water, followed by protein adsorption then cell interaction, among which is included MSCs. The surface macro scale is responsible for the interlock between bone and implant [45]. The micro scale can influence cell orientation [46] and potentially proliferation capacity and differentiation ability. The nano scale can influence cell-to-cell signalling [34], and can override biochemical cues [47]. Independent of the surface chemistry, the surface scale, namely micro and nano topography, has a significant effect on cell behaviour [48]. The cell cytoskeleton organisation is strongly affected by the orientation of the surface structures (i.e., physical roughness and topography), which stimulates cell contact guidance [49]. Contact guidance refers to the phenomenon where cells will adjust their orientation and align along nano-micro-groove-like patterns to grow. It was first observed by Harrison [50] in 1912, and the terminology first described by Weiss and Taylor [46] in 1945. Curtis and Wilkinson reviewed the materials (one being titanium surface oxides) and topographical structures which can effect cell behaviour, such as grooves, ridges, spikes and pits [51]. Research has evolved since, and now emphasis is placed on how cell geometric cues can direct cell differentiation, and in the process manipulate cells into square, rectangular and pentagon shapes [52].

The cell coverage was similar across the early attachment time points (Figure 5) which correlates with the proliferation data at Day 3 (Figure 6). Although the coverage was similar, the morphology was distinctive on the laser treated surfaces. At Day 14, proliferation results show that BM surface had the highest fluorescence intensity (significantly different from both laser treated surfaces), although at Day 14 the differentiation results show that BM had spindle shaped cells and approximately 60% coverage, while the laser treated groups first began to show morphological evidence of cluster formation, perhaps indicative of bone-like nodules, although additional in-depth analysis is required to further characterise the cell behaviour. It is clear that cell shape is a relevant parameter in the biomaterial design process, as a fundamental physiological feature of functional tissue [53]. Faster bone formation by laser surface treatment could be explained by the hypothesis that osteoblast precursors migrating into the pores of a rough surface reach confluence earlier within the enclosed space, cease proliferation and then differentiate [54]. Improvement of overall in vitro performance links to the surface roughness and topographical features, with these being the main indicators of osseointegration success [19,55,56]. The introduction of laser technology for titanium surface modification is feasible, and evidently beneficial for accelerating bone formation [57].

The antibacterial effect arising from the changes in surface chemistry of TNZT after laser treatment could apply to the attachment of MSCs, i.e., negatively impacting the MSCs attachment on laser treated surfaces. However, positive results of attachment and coverage of MSCs on the laser treated surfaces, at least comparable with that of the untreated (polished) surfaces, can still be observed in this study. It can be attributable to the size difference between bacteria and MSCs, namely 0.5–1 μm and 20–30 μm, respectively.

In the authors' recent study [16,36], bacteria were found to be insensitive to the micro-sized surface features, namely micro-ripples in the laser tracks [36], while the response of bacteria was very much dictated by the nano-sized features, i.e., nano-spiky features are effective to inhibit bacterial attachment and to kill the bacteria that attached [16]. Similarly, Puckett et al. found that certain nanometre sized titanium topographies may be useful for reducing bacteria adhesion while promoting bone tissue formation [58]. In contrast, the relatively large-sized MSCs, compared with the bacteria, are more sensitive to surface features in both the micro- and nano-sized range. It has been reported

by Chan et al. [59] that laser-induced surface ripples or patterns in micro size can encourage higher cell attachment of MSCs, leading to a higher cell coverage on the surfaces after laser treatment. Surface modification is emerging as a promising strategy for preventing biofilm formation on abiotic surfaces [60]. Implant success relies upon the surface inhibiting bacterial adherence and concomitantly promoting tissue growth.

To summarise, there is a competing process between effects caused by the laser-induced surface chemistry and micro-features, i.e., changes in surface chemistry after laser treatment could make the surface less hospitable to MSCs, whereas the micro-sized ripples (or physical features) can promote more cell attachment and coverage. However, at this stage it is still inconclusive as to how the laser-induced chemical or physical effects individually act on the response of MSCs, or which one is more dominant. A single surface feature effect cannot be studied in isolation from the others.

## 5. Conclusions

The beta titanium alloy Ti-35Nb-7Zr-6Ta coupled with fibre laser surface treatment in a high speed regime (ranging 100 and 200 mm/s) is a promising choice for load bearing implant applications. The surface roughness, topography and composition can be tailored by fibre laser treatment to improve in vitro mesenchymal stem cell attachment, proliferation and differentiation, as well as reducing bacterial attachment.

The major findings of this research are summarised below:

(1) Fibre laser treatment can be used to polish the TNZT surfaces in the high speed regime, with the scanning speed of 200 mm/s (or 12 m/min) being the most effective in this study;

(2) The laser treated samples exhibited surface homogenisation (or homogenous elemental distribution), and only showed beta phases after fibre laser treatment, namely β (110) and (200);

(3) Fibre laser treatment created a rougher (Ra value of BM was 199 nm, LT100 was 256 nm and LT200 was 232 nm) and spiky surface (Rsk > 0 and Rku > 3) with a homogenous elemental distribution and presence of TiN in the outmost oxide layer, which encouraged bone-like nodule formation and a bactericidal effect.

To summarise, the cell (attachment, proliferation and differentiation of MSCs) and bacterial culture (live/dead ratio of *S. aureus*) results indicate that LT200 is the optimal condition to treat the TNZT surface, giving the most desirable MSC responses and a significant reduction in bacterial adhesion.

**Supplementary Materials:** The following are available online at http://www.mdpi.com/2079-6412/9/3/186/s1.

**Author Contributions:** Conceptualization, C.L.D. and C.-W.C.; methodology, C.L.D., G.C.S. and L.C.; formal analysis, C.L.D., R.M., G.C.S., A.M. and C.-W.C.; investigation, C.L.D.; resources, R.M., G.C.S., S.K., L.C., S.M., A.M. and C.-W.C.; data curation, C.L.D. and C.-W.C.; writing—original draft preparation, C.L.D., G.C.S. and C.-W.C.; writing—review and editing, C.L.D., R.M.F., G.C.S., L.C., A.M. and C.-W.C.; visualization, C.L.D. and C.-W.C.; supervision, S.M., A.M. and C.-W.C.; project administration, C.-W.C.; funding acquisition, L.C., A.M. and C.-W.C.

**Funding:** The work described in this paper was supported by research grants from the Queen's University Belfast, awarded to C-WC and LC (D8201MAS, D8304PMY). The cell work was funded by Biotechnology and Biological Sciences Research Council (BBSRC) and British Heart Foundation (BHF) grants.

**Acknowledgments:** Anna Krasnodembskaya's group at the Centre for Experimental Medicine (Queen's University Belfast) for the provision of MSCs.

**Conflicts of Interest:** The authors declare no conflict of interest.

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
