# Peer review of "Fibre Laser Treatment of Beta TNZT Titanium Alloys for Load-Bearing Implant Applications: Effects of Surface Physical and Chemical Features on Mesenchymal Stem Cell Response and Staphylococcus aureus Bacterial Attachment"

_coatings, doi:10.3390/coatings9030186_

Round 1
Reviewer 1 Report
The topic of the manuscript is of high interest: laser surface modification for biomedical applications is a potential method to improve the development of specific topographies for the study of growth behavior of cells but, prior to publish, some minor modification to the paper are recommended.
In this sense, this research manuscript is relevant, well founded and discussed, and of interest to the audience of this journal. The introduction and literature review provides background information for understanding the methodology, which is appropriate and applied properly. Results and conclusions are correctly interpreted, clear and well explained
The abstract is concise and sufficient; Tables and figures are appropriate and complete.
Nevertheless, some observations, and changes are recommended.
This manuscript can be considered for publication after addressing the following comments:
1. Abstract and conclusions needs to address some values regarding your main findings as well.
2. Ti6Al4V is one of the most used alloys used for the development of biomedical devices, however, the use of TNZT alloy may improve some properties and features of the surface. Could you explain in economic terms a brief justification for the use of this alloy.
3. It is recommended to include the energy density value for each laser treatments.
Author Response
Please see below and the attached word file for our response to the reviewer 1's comments.
Reviewer 1 – Comments:
Abstract and conclusions needs to address some values regarding your main findings as well.
Response: Abstract and conclusions updated to include key values
ABSTRACT: Sample groups included untreated base metal (BM), laser treated at 100 mm/s (LT100) and 200 mm/s (LT200). The results demonstrated that laser surface treatment creates a rough (Ra value of BM is 199 nm, LT100 is 256 nm and LT200 is 232 nm) spiky surface (Rsk > 0 and Rku > 3) with homogenous elemental distribution and decreasing peak-to-peak distance between ripples (0.63 µm to 0.315 µm) as the scanning speed increases (p < 0.05), generating a surface with distinct micron and nano scale features. The improvement in cell spreading, formation of bone-like nodules (only seen on the laser treated samples) and subsequent four-fold reduction in bacterial attachment (p < 0.001) can be attributed to the features created through fibre laser treatment, making it an excellent choice for load bearing implant applications. Last but not least, the presence of TiN in the outermost surface oxide might also account for the improved biocompatibility and antibacterial performances of TNZT.
CONCLUSIONS: The major findings of this research are summarised below:
1) Fibre laser treatment can be used to polish the TNZT surfaces in the high speed regime, with the scanning speed of 200 mm/s (or 12 m/min) being the most effective in this study;
2) The laser treated samples exhibited surface homogenisation (or homogenous elemental distribution) and only showed beta phases after fibre laser treatment, namely β (110) and (200);
3) Fibre laser treatment created a rough (Ra value of BM is 199 nm, LT100 is 256 nm and LT200 is 232 nm) and spiky surface (Rsk > 0 and Rku > 3) with homogenous elemental distribution and presence of TiN in the outmost oxide layer which encouraged bone-like nodule formation and a bactericidal effect.
Ti6Al4V is one of the most used alloys used for the development of biomedical devices, however, the use of TNZT alloy may improve some properties and features of the surface. Could you explain in economic terms a brief justification for the use of this alloy?
Response: Introduction text updated to reflect the economic benefit of using TNZT over Ti-6Al-4 (inc. references to support statement)
INTRODUCTION: Other sources of pain after primary total hip replacement (THR) as recorded by the UK National Joint Registry (NJR), include adverse soft tissue reaction to particulates and to infection; each of which can be improved upon by combining better material selection in the form of a novel beta titanium alloy and surface modification using fibre laser treatment. In terms of the economic benefit, TNZT is potentially superior to currently used materials as it coupled with laser surface treatment can simultaneously decrease the most common indications for hip revision surgery, listed in the NJR reports1,26-29. TNZT use for hip stems may also have a longer life span beyond the current three-quarters of hip replacements that last 15-20 years30 before a revision surgery is required. A systematic review conducted suggested that just over half of hip replacements last 25 years30; as ever there is room for improvement.
References used as follows:
26. National Joint Registry. National Joint Registry 11th Annual Report 2014. (2014).
27. National Joint Registry. National Joint Registry 12th Annual Report 2015. (2015).
28. National Joint Registry. National Joint Registry 13th Annual Report 2016. (2016).
29. National Joint Registry. National Joint Registry 15th Annual Report 2018. (2018).
30. Evans, J. T. et al. How long does a hip replacement last? A systematic review and meta-analysis of case series and national registry reports with more than 15 years of follow-up. Lancet (London, England) 393, 647–654 (2019).
It is recommended to include the energy density value for each laser treatments.
Response: Experimental details updated to include size of laser treated area and laser energy. Calculations related to laser energy and intensity added as supplementary materials
LASER TREATMENT: The laser wavelength is 1064 nm. The samples were prepared using the following parameters: laser power 30 W, stand-off distance 1.5 mm, argon gas with 30 L/min flow rate and two different scanning speeds, 100 and 200 mm/s. Laser sample groups are denoted as LT100 and LT200 hereafter. The laser treated area of the surface was 6 mm2 in square shape. The laser energy at the two speeds are 1.8 and 0.9 J respectively (see supplementary material for calculations).
See supplementary material for calculations related to laser energy and intensity
Reviewer 2 Report
The work presented is very interesting and was conducted with a lot of attention to detail. The authors did a good job presenting the subject and the acquired data and discussing it. The writting is captivating and offers a clear prespective over the main conclusions. There are some Eglish mistakes that should be taken care with a new reading of the manuscript. After that I recommend the publication of this manuscript.
Author Response
Please see below and the attached word file for our response to reviewer 2's comments.
Reviewer 2 – Comments:
There are some English mistakes that should be taken care with a new reading of the manuscript.
Response: text updated where applicable to improve spelling and grammar.
INTRODUCTION: Selective material elemental composition of titanium (Ti) alloys coupled with surface treatment can be used to improve osseointegration and reduce bacterial adhesion.
Double spacing removed where applicable
Reviewer 3 Report
The manuscript present interesting and valuable results. In my opinion the paper will find broad scientific audience. I suggest to accept the manuscript in present form.
Author Response
Please find below and attached for our response to reviewer 3's comments:
Reviewer 3 – No changes suggested